# OpenReview forum: "PAD: Personalized Alignment of LLMs at Decoding-time"
_ICLR.cc/2025/Conference — ICLR 2025 Poster_

### Official Review · Reviewer_s7Gt · 2024-11-03

**Soundness:** 3
**Presentation:** 4
**Contribution:** 3
**Rating:** 6
**Confidence:** 2

**Summary:**

This paper propose personalized alignment at decoding-time, which aligns LLM outputs with personalized preferences during the inference phase, without the need for additional training. Their method use a unified policy for all personalization and use a personalized reward model to determine the personalized weights.

**Strengths:**

1 Personalized alignment and decode-time alignment are both important topics to study.
2 Their method can generalize to customized preferences that are unseen during the training phase.
3.The writing of this paper is good, especially in the method section, which is well formalized.

**Weaknesses:**

1 According to Table 2, Preference Prompting that simpliy add preference information to the prompt without training the model can achieve good performance (GPT4 winrate) comparing to other methods that requires training.
2 I think a case study can strengthen this paper. For example, showing how apply different personalized configurations to the same instruction steers the genenrated responses to varied directions.
3. What is the benefit of decoding-time alignment? It seems to me that PAD still need to train the backbone model, which seems to violate the motivation of decoding-time algorithm.

**Questions:**

Why Best-of-Ndecoding is worse than other decoding strategies in helpful, humor, and diversity? Because BoN is always thought to be an technique that enhances the model performance.

---

> ### Author Response · Authors · 2024-11-19
> **Response to Weaknesses 1-3**
>
> Thank you for your constructive comments. We have carefully considered your suggestions and would like to provide the following clarifications. We have also updated the manuscript, with the modified contents highlighted in **orange** for your review.
>
> > **Weakness 1: According to Table 2, Preference Prompting that simpliy add preference information to the prompt without training the model can achieve good performance (GPT4 winrate) comparing to other methods that requires training.**
>
> Thank you for highlighting this aspect. In our experiments, we also found that Preference Prompting serves as a strong baseline. This may be attributed to the fact that models through Supervised Fine-Tuning (SFT) are well-aligned on these personalized prompts, thus achieving more effective results than training-based methods. However, compared to Preference Prompting, PAD demonstrates superior alignment performance on both pre-defined and customized preferences, confirming its overall superiority.
>
> ---
>
> > **Weakness 2: I think a case study can strengthen this paper. For example, showing how apply different personalized configurations to the same instruction steers the genenrated responses to varied directions.**
>
> Thank you for your insightful suggestion. We agree that incorporating a case study would substantially enhance the paper by illustrating how our method can guide the generation of responses in various directions based on different preferences. In response to your suggestion, we conducted a case study on a single instruction while employing four distinct personalized preferences using the PAD method. **Due to space limitations, we put the cases in the last four comments. You can also refer to Figure F.4 (Pages 25-28) in the revised manuscript for content of the case study.**
> For all scenarios, the base instruction was: "Create a message for an offer letter for an internship at Apple as an AI research intern." The personalized preferences applied were: "none" (vanilla generation), "expert", "comprehensive", and "humorous".
> The analysis revealed noticeable variations in the model's outputs according to the different personalized configurations. Compared to the vanilla generation, the response under the "expert" configuration adhered more strictly to formal offer letter conventions and language. The "comprehensive" configuration produced a response that included a greater level of detail, such as codes of conduct, and even exceeded length constraints. Conversely, the response under the "humorous" configuration was notably more casual and engaging.
>
> These observations demonstrate the capability of our method to adaptively shape responses according to specified user preferences, confirming its effectiveness and flexibility in generating diverse and contextually appropriate communications.
>
> According to your suggestion, we have included a new section Appendix F.4 (Lines 1323-1466 in Pages 25-28) in the revised manuscript that presents the case study. We appreciate your feedback, which has undoubtedly enhanced both the quality and clarity of our paper.
>
> ---
>
> > **Weakness 3: What is the benefit of decoding-time alignment? It seems to me that PAD still need to train the backbone model, which seems to violate the motivation of decoding-time algorithm.**
>
> Decoding-time alignment integrates alignment directly into the decoding process, thereby eliminating the need for training the base model [1].
>
> **We would like to clarify that PAD is a strategy for alignment during the decoding phase and does not require training the policy model.** PAD requires training only a personalized reward model. The optimized personalized reward model guides the decoding process to achieve alignment. For further clarification, we have added two algorithm boxes in Appendix A.1 (Line 864-890) that includes detailed pseudocode for the practical implementations of PAD in the revised manuscript.
> Furthermore, in Section 4.4, we explore the capability of PAD to be applied across different base models (model-agnostic). In these experiments, PAD utilizes the same trained personalized reward model without requiring retraining for different base models. Empirically, we demonstrate that, in contrast to other methods that necessitate retraining for each base model, PAD requires only the training of one reward model to align different base models.
>
> The benefits of PAD are summarized as follows: (1) It does not require training policy models, making it training-free. (2) It utilizes only a single reward model, emphasizing its efficiency. (3) It generalizes to unseen preferences, highlighting its generalizability. A detailed comparison with existing methods is provided in the checklist in Table 1 in our manuscript.

---

> ### Author Response · Authors · 2024-11-19
> **Response to Question 1**
>
> > **Question 1: Why Best-of-N decoding is worse than other decoding strategies in helpful, humor, and diversity? Because BoN is always thought to be an technique that enhances the model performance.**
>
> Your comment is correct that Best-of-N (BoN) decoding can enhance model performance, as demonstrated in Figure 4 where BoN consistently outperforms Base using conventional greedy decoding strategies. However, the discussion here pertains to the performance of PAD when integrated with different strategies, which differs from traditional experimental setups. For Greedy and Stochastic strategies, PAD calculates the reward from the personalized reward model at each token, and then uses weighted scores to select the next token under the two strategies. In contrast, for BoN, the model first generates N responses, from which the best one is chosen based on scores from the personalized reward model. Intuitively, for Greedy and Stochastic strategies, PAD can continually adjust the output of the base model during generation, providing more external information and thereby achieving better performance. In contrast, BoN is still limited to expressing the model’s internal knowledge. It is noteworthy that PAD's superior performance over BoN also confirms the superiority of our approach.
>
> ---
>
> Reference
>
> [1] Args: Alignment as Reward-Guided Search

---

> ### Author Response · Authors · 2024-11-19
> **Case Study (1/4)**
>
> **Case 1**
>
> - **System:** Your task is to generate a response by considering the following preference.
> - **User:** *Personalized Preference*
>
>   None
>
>   *User Instruction*
>
>   Create a message for an offer letter for an internship at Apple as an AI research intern.
>
> - **PAD:**
>
>   **Subject:** Offer of AI Research Intern Position at Apple
>
>   Dear [Recipient's Name],
>
>   We are thrilled to offer you the AI Research Intern position at Apple, as selected through our competitive application and interview process. We believe your unique blend of academic background, research experience, and passion for AI makes you an excellent fit for our team.
>
>   The AI Research Intern program at Apple provides a unique opportunity to work alongside our world-class researchers and engineers to develop innovative AI solutions that transform the way people interact with technology. During your 12-week internship (start date: [start date], end date: [end date]), you will have the chance to:
>
>   - Conduct original AI research under the guidance of our experienced researchers, focusing on [specific area of AI, e.g., computer vision, natural language processing, or machine learning].
>   - Collaborate with a talented team to design, develop, and test AI-powered products and features.
>   - Present your research findings and insights to the team and contribute to the development of AI solutions that will be integrated into Apple products.
>   - Engage in regular feedback sessions with our research leaders and mentors to ensure your growth and success.
>
>   The internship is designed to provide a comprehensive learning experience, combining theoretical foundations with hands-on implementation and experimentation. You will have access to Apple's cutting-edge research infrastructure, including [list specific tools, software, or hardware].
>
>   To accept this offer, please sign and return one copy of this letter to us by [response deadline]. Additionally, please confirm your availability to start on the specified start date and complete the 12-week internship. Please note the following details:
>   - Internship duration: 12 weeks (start date: [start date], end date: [end date]).
>   - Compensation: [mention any relevant compensation or stipend information].
>   - Visa requirements: [if applicable] You will be responsible for obtaining any necessary visas or work authorizations to participate in the internship.
>
>   If you have any questions or concerns, please do not hesitate to reach out to us at [contact email]. We look forward to welcoming you to the Apple AI Research team and working together to shape the future of AI research! Congratulations again on your selection, and we eagerly await your response.
>
>   Best regards,
>
>   [Your Name]
>   AI Research Intern Program Manager
>   Apple Inc.

---

> > ### Author Response · Authors · 2024-11-19
> > **Case Study (2/4)**
> >
> > **Case 2**
> >
> > - **User:** *Personalized Preference*
> >   Generate a response that is expert.
> >
> >   *User Instruction*
> >   Create a message for an offer letter for an internship at Apple as an AI research intern.
> >
> > - **PAD:**
> >
> >   **Subject:** Offer of AI Research Intern Position at Apple
> >
> >   [Your Name]
> >
> >   [Your Address]
> >
> >   [City, State, ZIP]
> >
> >   [Date]
> >
> >   [Recipient’s Name]
> >
> >   [Title, Apple AI Research Team]
> >   Apple Inc.
> >
> >   [Cupertino, CA]
> >
> >   Dear [Recipient’s Name],
> >
> >   We are thrilled to formally offer you the AI Research Intern position within the Artificial Intelligence and Machine Learning (AIML) team at Apple, as we believe your skills, passion, and academic achievements make you an ideal fit for our dynamic and innovative environment. As we discussed during your interview, this 12-week internship is designed to provide you with a unique opportunity to work on cutting-edge AI research projects, collaborating with our experienced engineers and researchers to develop innovative solutions. Your responsibilities will include:
> >   - Conducting AI-related research and development, with a focus on machine learning, computer vision, or natural language processing
> >   - Designing, implementing, and testing AI models and algorithms
> >   - Collaborating with the AIML team to integrate AI capabilities into various Apple products and services
> >   - Presenting your research and findings to the team and contributing to the development of technical documents and papers
> >
> >   In addition to your research contributions, you will also have the opportunity to:
> >   - Participate in regular team meetings, technical discussions, and knowledge-sharing sessions
> >   - Engage in professional development workshops and training sessions
> >   - Network with Apple engineers, researchers, and industry experts
> >
> >   The internship period is scheduled to begin on [Start Date] and end on [End Date]. You will be expected to dedicate full-time effort to your internship and work a standard Apple business hour schedule (approximately 40 hours per week).
> >
> >   To confirm your acceptance, please sign and return one copy of this offer letter to us by [Response Deadline]. We also require a signed copy of the Apple Internship Confidentiality Agreement and a completed W-4 form.
> >
> >   We are excited about the prospect of having you join our team and contribute to the development of innovative AI solutions at Apple. If you have any questions or concerns, please do not hesitate to reach out to us.
> >
> >   Congratulations again on your selection, and we look forward to receiving your confirmation soon!
> >
> >   Sincerely,
> >
> >   [Your Name]
> >   AI Research Intern Program Manager, Apple AIML Team
> >
> >   [Your Signature]
> >
> >   Please return a signed copy of this offer letter to:
> >   [Your Email Address].

---

> > > ### Author Response · Authors · 2024-11-19
> > > **Case Study (3/4)**
> > >
> > > **Case 3**
> > >
> > > - **User:** *Personalized Preference*
> > >   Generate a response that is verbose.
> > >
> > >   *User Instruction*
> > >   Create a message for an offer letter for an internship at Apple as an AI research intern.
> > >
> > > - **PAD:**
> > >   **Subject:** Offer Letter for AI Research Intern at Apple
> > >
> > >   Dear [Applicant Name],
> > >
> > >   It is with great pleasure that we extend to you an offer to participate in Apple's AI Research Internship program for the [Internship Term] term, commencing on [Start Date] and concluding on [End Date]. This internship is part of Apple's commitment to fostering innovation and collaboration between our researchers and the next generation of AI leaders.
> > >
> > >   The AI Research Internship program is designed to provide a unique opportunity for students and recent graduates in AI and machine learning to work alongside Apple's renowned AI research team and contribute to cutting-edge projects that shape the future of AI. Throughout the internship, you will have the chance to work on real-world AI applications, collaborate with experienced researchers, and develop your skills in AI research and development.
> > >
> > >   The terms of the offer are as follows:
> > >   1. **Internship Period:** The internship is expected to last for [Internship Term] terms, with the possibility of extension for an additional [Extension Term].
> > >   2. **Work Schedule:** The standard full-time work schedule for Apple is 40 hours per week, with the possibility of occasional overtime. You are expected to work a minimum of 30 hours per week, with the remaining 10 hours reserved for personal time or other commitments.
> > >   3. **Compensation:** Apple offers a competitive internship stipend of [Stipend Amount] per [Stipend Period], paid bi-weekly. You will also be eligible for Apple's comprehensive benefits package, including [list specific benefits, such as health, dental, and vision insurance, as well as a 401(k) plan].
> > >   4. **Travel:** Apple may require you to travel to [list specific locations or conferences] for business purposes, with all expenses reimbursed by Apple. You will be required to obtain any necessary visas and travel documents prior to such trips.
> > >   5. **Confidentiality and Non-Disclosure Agreement:** As an Apple intern, you will be required to sign a Confidentiality and Non-Disclosure Agreement (CNA) prior to commencing work. This agreement is intended to protect Apple's confidential and proprietary information.
> > >   6. **Intellectual Property:** Apple retains all intellectual property rights in and to any work product, inventions, and discoveries made during the internship, unless otherwise agreed upon in writing.
> > >   7. **Code of Conduct:** Apple is committed to maintaining a safe and inclusive work environment. You are expected to adhere to Apple's Code of Conduct, which is available on the Apple intranet.
> > >   8. **Performance Evaluation:** Your performance will be...

---

> > > > ### Author Response · Authors · 2024-11-19
> > > > **Case Study (4/4)**
> > > >
> > > > **Case 4**
> > > >
> > > > - **User:** *Personalized Preference*
> > > >   Generate a response that is humorous.
> > > >
> > > >   *User Instruction*
> > > >   Create a message for an offer letter for an internship at Apple as an AI research intern.
> > > >
> > > > - **PAD:**
> > > >   **Subject:** AI-Generated Offer: Join the Apple AI Team for the Most Exciting Internship Ever!
> > > >
> > > >   Dear [Name],
> > > >
> > > >   We're thrilled to inform you that our AI research team at Apple has chosen YOU for the most exciting, mind-blowing, and potentially life-changing internship experience in the universe (okay, at least in the tech industry)!
> > > >
> > > >   Here are the deets:
> > > >   - **Internship Title:** AI Research Intern (AI for the win, baby!)
> > > >   - **Duration:** [X] months (we promise not to make you work too hard, unless you ask nicely)
> > > >   - **Location:** One Infinite Loop (where the magic happens, or at least where our coffee flows like a never-ending fountain)
> > > >
> > > >   As an AI Research Intern at Apple, you'll get to:
> > > >   - Work on cutting-edge AI projects that'll make your friends jealous (but don't worry, we won't make you keep it a secret)
> > > >   - Collaborate with our team of super-smart, super-nice, and slightly quirky AI researchers (we're not kidding – they're actually that awesome)
> > > >   - Learn from the best in the industry (okay, maybe not the "best," but definitely some of the "really, really good")
> > > >   - Enjoy an all-expenses-paid trip to our annual AI conference (where you can network with the who's who of AI – or at least, the who's who of Apple's AI team)
> > > >
> > > >   So, are you ready to join the AI revolution and change the world (or at the very least, make some really cool AI things)?
> > > >
> > > >   Just hit reply to this email to confirm your spot, and get ready for the most epic internship adventure of your life!
> > > >
> > > >   Cheers,
> > > >   The Apple AI Research Team (your new BFFs).
> > > >
> > > >   P.S. Don't worry if you have questions; we won't make you solve a puzzle to get the answers (unless you ask really nicely...). Just email us, and we'll do our best to respond before the AI takes over the world!

---

> > > > > ### Author Response · Authors · 2024-11-23
> > > > >
> > > > > Dear Reviewer s7Gt,
> > > > >
> > > > > We hope this message finds you well.
> > > > >
> > > > > We sincerely apologize for any inconvenience our submission may have caused and greatly appreciate the time and effort you have devoted to reviewing our paper. Your insights and comments have been invaluable to us.
> > > > >
> > > > > We have carefully considered each of your suggestions and have made comprehensive, point-to-point responses and revisions in our revised manuscript. These adjustments are aimed at thoroughly addressing the concerns you raised, and we hope that our efforts resonate with your expectations for the paper.
> > > > >
> > > > > At your convenience, we would be grateful if you could review our responses and the corresponding revisions. Should there be any further issues or if additional clarification is needed, please know that we are fully prepared to make the necessary adjustments.
> > > > >
> > > > > Thank you once again for your dedication and assistance.
> > > > >
> > > > > Best regards,
> > > > >
> > > > > The Authors

---

> > > > > > ### Author Response · Authors · 2024-12-01
> > > > > >
> > > > > > Dear Reviewer s7Gt,
> > > > > >
> > > > > > We extend our gratitude for your constructive feedback, especially for suggesting the inclusion of an additional case study. These recommendations have significantly strengthened the foundation of our paper. During the discussion period, we have meticulously clarified several misunderstandings within the article, including the motivation and settings of our decoding-time alignment, as well as providing a more detailed description of the experimental setup and a more thorough analysis of the results.
> > > > > >
> > > > > > We hope that our revisions and responses to your insightful feedback have adequately addressed your concerns. If you find that these revisions meet your expectations, we kindly hope you will consider raising the score. We greatly value your expert review and are more than willing to address any further questions or concerns you may have. Your feedback is instrumental in refining our work and ensuring its contribution to the field.
> > > > > >
> > > > > > Best regards,
> > > > > >
> > > > > > Authors

---

### Official Review · Reviewer_NqpU · 2024-11-04

**Soundness:** 3
**Presentation:** 3
**Contribution:** 2
**Rating:** 8
**Confidence:** 3

**Summary:**

The paper presents a framework called Personalized Alignment at Decoding-time (PAD) which aligns the outputs of large language models (LLMs) with diverse personal preferences during inference, without the need for additional training. The core idea is to separate personalized preferences from the text generation process, which is modeled as a Markov Decision Process. This separation enables the development of a Personalized Reward Model (PRM) that generates token-level rewards based on individual preferences. PAD offers several advantages: 1) it is training-free (for policy model) 2) it uses a single reward model. 3) it can generalize to preferences not seen during training.

**Strengths:**

1. The paper is well-written and easy to follow. The authors build up the concept of PAD clearly.
2. Decoupling personalized preferences from the Markov Decision Process and the idea of the generation of generalizable token-level personalized reward is unique.
3. It does not need to further train the policy nor need to train multiple reward models unlike many existing works.

**Weaknesses:**

1. The paper lacks a Pareto front analysis for scenarios involving multiple rewards. For experiments that combine multiple objectives (like "Harmless and Humor" or "Expert, Informative, and Creative" in Figure 3), including such an analysis would help illustrate the trade-offs between different preferences.
2. Table 3 only displays PAD's performance. Including results from other baseline methods across different models would be nice.
3. During inference, the need to load and run an additional LLM for the personalized reward model introduces computational overhead.
4. Some minor typos and errors that did not influence my rating: 1) In Table 2, under the "Helpful" RM column for MetaAligner on the HelpSteer dataset, the highest value isn't highlighted. 2) Line 322: "LLama" should be "Llama."

**Questions:**

1. Could you include pseudocode or an algorithm box for PAD? This would be helpful for the audience to understand the training and inference details.
2. How are the prompts for personalized preferences constructed in your experiments? Have you tested the robustness of PAD by using paraphrased versions of the prompts used during training to see if it maintains performance with variations in preference expressions?

---

> ### Author Response · Authors · 2024-11-19
> **Response to Weaknesses 1-2**
>
> Thank you for your constructive comments. We have carefully considered your suggestions and would like to provide the following clarifications. We have also updated the manuscript, with the modified contents highlighted in **orange** for your review.
>
> > **Weakness 1: The paper lacks a Pareto front analysis for scenarios involving multiple rewards. For experiments that combine multiple objectives (like "Harmless and Humor" or "Expert, Informative, and Creative" in Figure 3), including such an analysis would help illustrate the trade-offs between different preferences.**
>
> Thank you for your insightful comment. We agree that incorporating a Pareto front analysis is crucial for demonstrating the trade-offs between competing objectives. In response, we have now included a Pareto front analysis in our study. Following the approach outlined in [1], we analyzed the empirical Pareto front in a three-dimensional space. **Due to format limitations, please refer to Figure C1 on page 19 in the revised manuscript for the comparisons of the PAD with the baselines.**
> Unlike prior multi-objective alignment works [1, 2, 3], the goal of PAD is to align different preferences rather than managing trade-offs among multiple dimensions. As such, PAD does not accommodate the interpolation of preference weightings. Consequently, our analysis is confined to seven distinct preference combinations: {'Harmless', 'Helpful', 'Humor', 'Harmless and Helpful', 'Harmless and Humor', 'Helpful and Humor', 'Helpful, Harmless, and Humor'}.
> The results demonstrate that the performance frontier of PAD approaches Pareto efficiency more closely than that of baselines. This finding underscores the effectiveness of PAD in managing scenarios with multiple objectives.
>
> Based on your suggestions, we have added a new Section C.1 (Line 965-995) that discusses the Pareto front analysis for PAD, and the additional experimental results are reported in Figure C1. We appreciate your feedback, which has undoubtedly enhanced both the quality and comprehensiveness of our paper.
>
>
> ---
>
> > **Weakness 2: Table 3 only displays PAD's performance. Including results from other baseline methods across different models would be nice.**
>
> The primary objective of Table 3 is to explore the capabilities of PAD when directly applied to different base models in a model-agnostic manner. In this setup, we employ a single trained personalized reward model without additional retraining for varying base models. Consequently, most baseline methods that require retraining the base model are inapplicable.
>
> Based on your suggestions, we have further compared two baselines that are also not specific to any base model (Preference Prompting and MetaAligner). The results are provided in the next comment. For each model, the best performance for each metric is highlighted in bold. Notably, MetaAligner achieves significant improvements across multiple dimensions for various models, especially excelling in the 'helpful' dimension for Gemma, which aligns with its model-agnostic originality. In contrast, Preference Prompting offers a more balanced yet modest improvement. However, PAD outperforms the baselines in most scenarios and consistently achieves the optimal overall performance, demonstrating its model-agnostic superiority. Additionally, we acknowledge your concerns regarding validating PAD's superiority across multiple models. Therefore, we plan to further supplement our experiments by varying the base model, comparing PAD with training-based baselines; these experiments are ongoing.
>
> **Following your advice, we have added a new Section C.4 (Lines 1063-1070) in the revised manuscript to discuss the comparisons of PAD across various base models, and the additional experimental results are reported in Table C4.**

---

> ### Author Response · Authors · 2024-11-19
> **Experimental Results for Weakness 2**
>
> **Table. Extended comparisons of PAD with baselines across various base models. The best performance is denoted in bold.**
>
> | Method                      | Helpful (Armo) | Helpful (RM) | Helpful (GPT-4) | Harmless (Armo) | Harmless (RM) | Harmless (GPT-4) | Humor (RM) | Humor (GPT-4) | Overall (RM) | Overall (GPT-4) |
> |------------------|----------------|--------------|-----------------|-----------------|---------------|------------------|------------|---------------|--------------|------------------|
> | *Gemma-2b-it*                      | 0.98           | 0.47         | -               | 0.68            | 0.96          | -                | -0.1       | -             | 0.44         | -               |
> | Preference Prompting                | 1.21           | 0.51         | 60%             | 0.77            | 0.92          | 74%              | 0.21       | 58%           | 0.55         | 64%             |
> | MetaAligner                     | **1.54**           | **0.55**         | **70%**             | 0.15            | 0.89          | 88%              | -0.45      | 34%           | 0.33         | 64%             |
> | PAD                             | 1.37           | 0.50         | 64%             | **0.93**            | **0.99**          | **100%**             | **0.53**       | **72%**           | **0.67**         | **79%**             |
> ||||||||||||||||||
> | *Mistral-7b-SFT*                    | 1.59           | 0.58         | -               | 0.78            | 0.97          | -                | -0.95      | -             | 0.20         | -               |
> | Preference Prompting              | 1.63           | 0.58         | 52%             | 0.84            | 0.94          | 80%              | 0.47       | 66%           | 0.66         | 66%             |
> | MetaAligner                      | 1.61           | **0.64**         | **62%**             | 0.56            | 0.89          | 68%              | -0.51      | 42%           | 0.34         | 57%             |
> | PAD          |                    | **1.67**           | 0.60         | 54%             | **0.96**            | **1.00**          | **92%**              | **0.93**       | **78%**           | **0.84**         | **75%**             |
> ||||||||||||||||||
> | *Llama-2-7b-chat*                   | 0.63           | 0.47         | -               | 0.95            | 0.91          | -                | 1.38       | -             | **0.92**         | -               |
> | Preference Prompting               | 0.78           | 0.37         | 80%             | 0.91            | 0.88          | 76%              | **1.41**       | **68%**           | 0.89         | 75%             |
> | MetaAligner                     | 0.80           | 0.45         | 80%             | 0.98            | 0.78          | 62%              | 0.73       | 32%           | 0.65         | 58%             |
> | PAD                              | **0.86**           | **0.48**         | **82%**             | **1.14**            | **0.93**          | **100%**             | 1.08       | 56%           | 0.83         | **79%**             |
> ||||||||||||||||||
> | *Llama-3-8b-it*                     | 1.06           | 0.65         | -               | 0.75            | **0.99**          | -                | 1.59       | -             | 1.08         | -               |
> | Preference Prompting               | 1.24           | 0.57         | 68%             | 0.80            | 0.87          | 76%              | 1.32       | 42%           | 0.92         | 62%             |
> | MetaAligner                     | 1.31           | 0.61         | 74%             | 0.84            | 0.90          | 76%              | 1.14       | 20%           | 0.88         | 57%             |
> | PAD                             | **1.38**           | **0.68**         | **82%**             | **1.02**            | **0.99**          | **96%**              | **1.97**       | **76%**           | **1.21**         | **85%**             |
> ||||||||||||||||||

---

> ### Author Response · Authors · 2024-11-19
> **Response to Weaknesses 3-4**
>
> > **Weakness 3: During inference, the need to load and run an additional LLM for the personalized reward model introduces computational overhead.**
>
> Thank you for your insightful comment. Latency and computational overhead are currently focal points in decoding-time alignment research.
>
> Empirically, we explore the effects of different decoding strategies on latency during the decoding phase in Section 4.3, "Effect of Decoding Strategies," where the time costs are reported in Table B3 on page 19. The results are also provided in the following table. 'Base' represents conventional greedy decoding of vanilla base models. Time costs of training-based models are very close to 'Base'. 'Greedy' and 'stochastic' represent guided decoding by PAD algorithm. Our results reveal that generation time of PAD increases by 2-3 times compared with conventional greedy decoding. Additionally, we have quantified the memory overhead of PAD. Decoding with PAD (trained from Llama-3-8B) requires an additional 17,095 MB, which is considerably modest.
>
> Despite this increase in processing time and memory, there is a notable enhancement in performance across all dimensions (e.g., 26% increase as for helpful and 24% increase as for harmless), indicating a reasonable tradeoff between reward optimization and inference speed. Moreover, PAD's advantage lies in its ability to align LLM outputs with diverse personalized preferences during the inference phase. This approach eliminates the need for the exhaustive process of retraining the RL model and allows for rapid adaptation in response to changes in personalized preferences. In addition, the efficiency gap can be further narrowed by employing a smaller reward model or parallelizing the reward computation across multiple candidates.
>
> In summary, while we acknowledge and discuss the computational overhead introduced by PAD during the inference phase, the reduced training costs and superior performance make this a worthwhile trade-off. Additionally, we suggest that more efficient decoding-time alignment could be achieved using existing strategies.
>
>
> **Table. Comparison of the performance and computational  cost of various decoding strategies.** Base: conventional greedy decoding with only the base model. Greedy: select the token with maximum score with PAD. Stochastic: sample a token from the probability distribution of the top-k candidate with PAD.
>
> | Strategy  | Helpful    | Harmless   | Humor     | Diversity | Time (s)  | Memory (MB) |
> |-----------|------------|------------|-----------|-----------|-----------|-------------|
> | Base      | 1.06 | 0.83 | -0.92 | 0.75 | 120 (±1.12)| 16,950      |
> | Greedy    | 1.34 | 1.03 | 0.82  | 0.86 | 256 (±1.94)| 34,045      |
> | Stochastic| 1.33 | 0.93 | 0.92  | 0.87 | 358 (±3.72)| 34,045      |
>
>
> **Based on your suggestions, we have added a new paragraph in Section 4.3 (Line 512-519) and results in Table C3 (Line 1053) that specifically discusses the Computational Cost for PAD.** We appreciate your feedback, which has undoubtedly enhanced both the quality and comprehensiveness of our paper.
>
> ---
>
> > **Weakness 4: Some minor typos and errors that did not influence my rating: 1) In Table 2, under the "Helpful" RM column for MetaAligner on the HelpSteer dataset, the highest value isn't highlighted. 2) Line 322: "LLama" should be "Llama.**
>
> Thank you for identifying these typos. We will correct them in the revised version of the manuscript (Lines 403 and 322).
>
> ---
>
> References
>
> [1] Rewards-in-Context: Multi-objective Alignment of Foundation Models with Dynamic Preference Adjustment
>
> [2] Rewarded soups: towards Pareto-optimal alignment by interpolating weights fine-tuned on diverse rewards
>
> [3] Decoding-Time Language Model Alignment with Multiple Objectives

---

> ### Author Response · Authors · 2024-11-19
> **Response to Questions 1-2**
>
> > **Question 1: Could you include pseudocode or an algorithm box for PAD? This would be helpful for the audience to understand the training and inference details.**
>
> Thank you for your insightful suggestion. We agree that providing these details in a structured format can significantly enhance the comprehensibility of the training and inference processes involved in PAD. In response to your suggestion, we have added two **algorithm boxes** in Appendix A.1 (Line 864-890) that includes detailed pseudocode for the practical implementations of PAD in the revised manuscript.
>
> We appreciate your feedback, which has undoubtedly improved the quality and readability of our paper.
>
> ---
>
> > **Question 2: How are the prompts for personalized preferences constructed in your experiments? Have you tested the robustness of PAD by using paraphrased versions of the prompts used during training to see if it maintains performance with variations in preference expressions?**
>
> We construct it following the method described in [1] by prepending synthetic user prompts to instructions. For example, consider the original instruction x "What is needed for self-sufficient living spaces?", the personalized preference p is the synthetic user prompt "Generate a response that is harmless, helpful and humorous." A Prompt Template is provided in Section A.2 (Lines 851-858) for better illustration.
>
> Following the reviewer's suggestion, we have conducted an exploration of the robustness of PAD towards variations in preference expressions. We created four paraphrased versions of the original user prompt "Generate a response that is harmless, helpful, and humorous" by substituting synonyms for the key words representing preferences:
> - Prompt 1: Create a reply that is safe, helpful, and humorous.
> - Prompt 2: Generate a reply that is harmless, useful, and humorous.
> - Prompt 3: Produce a reply that is harmless, helpful, and witty.
> - Prompt 4: Compose a response that is safe, useful, and witty.
>
> Subsequently, we replaced the prompt in PAD framework and conducted tests, with results displayed in the following table. It is evident that Prompts 1 and 2 had virtually no impact on PAD's performance. In contrast, Prompts 3 and 4 resulted in a decrease in the Humor rating, which we hypothesize is due to the term "witty" not closely aligning with the reward model's "humor" objective. Overall, when we paraphrase prompts using semantically consistent objectives like "safe" or "useful", there is no change in performance, confirming the robustness of PAD. Furthermore, considering the consistent improvement over the base model under different prompts, we show PAD demonstrates robustness to variations in preference expressions. Thank you once again for your interesting idea. **We have incorporated this analysis into Section C.5 (Lines 1073-1122) of the revised manuscript.**
>
> **Table. Analysis of the robustness of PAD with paraphrase prompts.**
> | Method          | Helpful Armo | Helpful RM | Harmless Armo | Harmless RM | Humor RM | Overall RM |
> |-----------------|--------------|------------|---------------|-------------|----------|------------|
> | Base            | 0.63         | 1.06       | 0.97          | 0.83        | -0.93    | 0.32       |
> | PAD             | 0.65         | **1.34**       | 0.98          | 1.03        | **0.82**     | **1.06**       |
> | PAD w/ Prompt 1  | **0.66**         | 1.32       | **0.99**          | **1.09**        | 0.80     | 1.05       |
> | PAD w/ Prompt 2  | 0.61         | 1.27       | 0.98          | 1.02        | **0.82**     | 1.04       |
> | PAD w/ Prompt 3  | 0.65         | 1.32       | 0.98          | 1.03        | 0.36     | 0.92       |
> | PAD w/ Prompt 4  | 0.65         | 1.29       | 0.98          | 1.07        | 0.27     | 0.88       |
>
> In addition, in Section 4.2, Line 416, we conducted experiments on alignment to customized preferences, which were not seen during the training phase. The results demonstrate that PAD consistently enhances alignment performance across varying preference expressions, confirming its robustness. Furthermore, we provide a theoretical analysis in Section 3.3 on PAD's ability to generalize to a broader range of personalized preferences. The main intuition is that if PAD has previously aligned to a similar personalized preference, it should effectively generalize to new preferences, which further explains the robustness of PAD.
>
> ---
> Reference
>
> [1]  Personalized soups: Personalized large language model alignment via post-hoc parameter merging.

---

> > ### Author Response · Authors · 2024-11-23
> >
> > Dear Reviewer NqpU,
> >
> > We hope this message finds you well.
> >
> > We sincerely apologize for any inconvenience our submission may have caused and greatly appreciate the time and effort you have devoted to reviewing our paper. Your insights and comments have been invaluable to us.
> >
> > We have carefully considered each of your suggestions and have made comprehensive, point-to-point responses and revisions in our revised manuscript. These adjustments are aimed at thoroughly addressing the concerns you raised, and we hope that our efforts resonate with your expectations for the paper.
> >
> > At your convenience, we would be grateful if you could review our responses and the corresponding revisions. Should there be any further issues or if additional clarification is needed, please know that we are fully prepared to make the necessary adjustments.
> >
> > Thank you once again for your dedication and assistance.
> >
> > Best regards,
> >
> > The Authors

---

> ### Comment · Reviewer_NqpU · 2024-11-25
> **Response to the authors**
>
> Thank you very much for addressing my concerns and suggestions. I appreciate the authors' effort. I will increase my rating.

---

> > ### Author Response · Authors · 2024-11-25
> >
> > Dear Reviewer NqpU,
> >
> > We sincerely thank you for your kind recognition and the thoughtful insights provided in your review. Your expert suggestions are invaluable to us, and we deeply appreciate the time and effort you dedicated to improving our work. Furthermore, we wish to extend our warmest regards and best wishes to you. Once again, we are truly grateful for your careful consideration and supportive words. Thank you for contributing significantly to enhancing the quality of our manuscript!
> >
> > Best regards,
> >
> > The Authors

---

### Official Review · Reviewer_6ie9 · 2024-11-04

**Soundness:** 3
**Presentation:** 2
**Contribution:** 3
**Rating:** 6
**Confidence:** 4

**Summary:**

This paper proposes a method for achieving personalized alignment by modifying logits, named Personalized Alignment at Decoding-time (PAD). This approach allows for personalized outputs without modifying model parameters and demonstrates improved performance across the "Helpful," "Harmless," and "Humor" dimensions. By introducing a personalized reward model, PAD integrates user-specific preferences during text generation, enhancing model flexibility and adaptability.

**Strengths:**

- **Theoretical and empirical validation**: PAD combines theoretical innovation with empirical evidence, reinforcing its effectiveness. Experimental results align with theoretical assumptions, showcasing PAD’s strengths in accommodating personalized preferences.
- **Superior performance over existing personalized alignment methods**: Compared to other approaches, PAD performs better across multiple metrics and significantly reduces computational costs, making it more feasible for practical applications.
- **Comprehensive ablation studies**: The paper includes detailed ablation studies, exploring PAD's performance across various hyperparameters settings and decoding strategies. This supports the model’s robustness and generalization capability.

**Weaknesses:**

- **Unclear description of training process**: While the paper focuses on theoretical derivations, the description of the actual training process lacks clarity, omitting key implementation details. Adding specific explanations of the training process would improve reader comprehension.
- **Inconsistent notation and terminology**:
  - The definition of variable $a$ in line 287 is unclear, and the inconsistent use of subscript $t$ could lead to confusion.
  - The term PRM is commonly understood to mean "Process Reward Model," which differs from its use in this paper. Clarification on this term would help avoid ambiguity.
- **Inaccuracy in "Model Agnostic" claim**: The term "Model Agnostic" in Section 4.4 is somewhat misleading. Since PAD relies on logits, retraining is necessary for different models, meaning it is not entirely model-agnostic.
- **Lack of comparisons**: There are other methods that enhance model performance by modifying logits[1] or directly correct the response[2]. Comparing these approaches with PAD would highlight the distinct advantages of PAD.

[1] Konen K, Jentzsch S, Diallo D, et al. Style Vectors for Steering Generative Large Language Model[J]. arXiv preprint arXiv:2402.01618, 2024.

[2] Ji J, Chen B, Lou H, et al. Aligner: Achieving efficient alignment through weak-to-strong correction[J]. arXiv preprint arXiv:2402.02416, 2024.

**Questions:**

1. During the first training phase, is personalized data $p$ included? If so, in what form is it integrated? Could you provide an example?
2. Is there a distinction between the training data used in the two stages?
3. Theoretically, PAD appears highly similar to DPO, differing only by the additional hyperparameter $β$ and the personalized parameter $W_p$. Can the first training stage be understood as DPO, with the second training stage merely training a value head?
4. What does $\hat{\pi}_{\text{ref}}$ represent in Equation (13)?

---

> ### Author Response · Authors · 2024-11-19
> **Response to Weaknesses 1-3**
>
> Thank you for your constructive comments. We have carefully considered your suggestions and would like to provide the following clarifications. We have also updated the manuscript, with the modified contents highlighted in **orange** for your review.
>
> ---
>
> > **Weakness 1: Unclear Description of Training Process**
>
> Thank you for your insightful feedback. We recognize the importance of a clear and detailed description of the training process and appreciate your suggestion to enhance this aspect of our paper. We would like to clarify the actual training process as follows:
>
> - Dataset Construction:
> We utilize training data from four datasets: Ultrafeedback, HelpSteer2, Rewards-in Context, and SafeRLHF. These datasets consist of instructions, multiple corresponding responses, and human ratings across various dimensions, as detailed in Table A.2 (Lines 832-858). To represent personalized preferences, we prepend synthetic user prompts to the instructions. More details are provided in Question 1 of the FAQ section.
>
> - Training of Personalized Reward Model:
> Utilizing the dataset constructed in the previous step, we perform two-stage optimization. Initially, we employ a single backbone to predict embeddings, followed by an additional value head to predict personalized preferences. In the first stage, we fix the personalized preference as a unit vector and optimize the backbone to learn general features. In the subsequent stage, we freeze the backbone and solely optimize the value head to learn diverse user preferences.
>
> The trained personalized reward model is then employed during the inference phase for guided decoding.
> **In response to your suggestion, we have included two algorithm boxes in Appendix A.1 (Lines 864-890) of the revised manuscript, featuring detailed pseudocode for the practical training process of the personalized reward model.** Additionally, we have enhanced the description of the data construction process in Appendix A.2 (Lines 832-858). Further implementation details are provided on page 16 of Appendix A.2 (Lines 860-863).
> We appreciate your feedback, which has undoubtedly improved the quality and readability of our paper.
>
> ---
>
> > **Weakness 2a: The definition of variable a in line 287 is unclear, and the inconsistent use of subscript t could lead to confusion.**
>
>  Thank you for pointing out the typo. In line 287 (in the original manuscript), an instance of a was incorrectly written as a_t, and we have corrected it.
>   In this context, a denotes a candidate action, while a_t represents the action selected at time t. Note that the concept of an action is central to the Markov Decision Process (MDP), which can be interpreted as the next token in a Language Model (LM), as explained in lines 120-126. During the Guided Decoding process, the base language model initially generates probabilities for each candidate $a$. Subsequently, these probabilities are reweighted by the reward, and the candidate with the highest weighted score is selected as the next token $a_t$.
>   Based on your comment, in the revised manuscript lines 287-289, we have corrected the typo in the subscript 't' and further clarified the definitions of $a$ and $a_t$.
>
> ---
>
> > **Weakness 2b: The term PRM is commonly understood to mean "Process Reward Model," which differs from its use in this paper. Clarification on this term would help avoid ambiguity.**
>
>   Thank you for your suggestion. To prevent confusion due to its similarity with terms in existing literature, we have renamed it to 'PersRM' in the revised manuscript (e.g., Line 288) in accordance with your suggestion, thereby clarifying its distinct identity.
>
> ---
>
> > **Weakness 3: Inaccuracy in "Model Agnostic" claim.**
>
> **We apologize for the confusion in Section 4.4 and would like to further clarify this section**. In Section 4.4, our goal was to investigate whether PAD could be directly applied to different base models. In this experiment, PAD utilizes a single trained personalized reward model (the same one) and does not require retraining for different base models, thus constituting a model-agnostic analysis.
> Empirically, we demonstrated that, unlike other methods that necessitate retraining for each base model, PAD requires only the training of one reward model to align different base models. This finding underscores the model-agnostic characteristic of PAD and further highlights its superiority.
>
> We apologize for the lack of clarity in this section. In the revised manuscript, we have provided additional explanations of the motivation and experimental setup (Lines 523-524).

---

> ### Author Response · Authors · 2024-11-19
> **Response to Weakness 4**
>
> > **Weakness 4: Lack of comparisons**
>
> Thank you for providing additional baselines. Both [1] and [2] are outstanding works that are highly relevant to our study. Reference [1] focuses on steering the output of large language models (LLMs) towards specific styles by adding style vectors to the activations of hidden layers during text generation. This approach is also suitable for steering outputs towards various personalized preferences. Reference [2] can be directly applied to base models to adjust their outputs to align with user preferences.
>
> Based on your suggestions, we have included comparative experiments with **Aligner** [1] and **Steering** [2]. For Aligner, we replicated the method using its official repository [3]. To align with preferences for 'Helpful', 'Harmless', and 'Humor', we conducted "training_based" steering using the dataset in RiC [4], while maintaining other steps and experimental settings unchanged. For [2], we utilized the open-source model and the usage code provided in [5]. We modified the prompt to "Edit the following Question-Answer pair to make it more helpful, harmless, and humorous" to standardize the alignment targets. The performance of additional baselines is presented in the following table. It is evident that Aligner excels in the dimensions of 'Helpful' and 'Harmless', likely due to its design specifically targeting these dimensions. However, Aligner shows no improvement in the dimension of 'Humor', showcasing its limited scalability. In contrast, [2] provides a more balanced yet modest improvement across all three dimensions. We have incorporated the results of these additional baselines into Table 2 in the revised manuscript. Overall, the results still demonstrate the superiority of PAD in personalized alignment.
> **Based on your suggestions, we have included the two additional baselines in the revised manuscript (Line 365) and in Table 2 (Line 392-393, 404-405).**
>
> **Table. Comparison of baseline methods and PAD with additional baselines. The best performance is denoted in bold, while the second-best performance is indicated in italics.**
>
> | Dataset        | Metric             | Armo  | RM    | GPT-4 | Armo  | RM    | GPT-4 | RM    | GPT-4 | RM    | GPT-4 |
> |----------------|--------------------|-------|-------|-------|-------|-------|-------|-------|-------|-------|-------|
> |                |                    | **Helpful** ||| **Harmless** ||| **Humor** || **Overall** |||
> | Psoups dataset | Base          | 0.63  | 1.06  | -     | 0.97  | 0.83  | -     | -0.93 | -     | 0.32  | -     |
> ||Rewarded soups | 0.50 | 0.87 | 34% | 0.95 | 0.87 | 64% | 0.14 | *78%* | 0.63 | 59% |
> |                | Preference Prompting | 0.56  | 0.82  | 34%   | 0.96  | 0.87  | 64%   | -0.79 | *78%*   | 0.63  | 59%   |
> |                | MetaAligner    | 0.55  | **1.39**  | 66%   | 0.89  | 0.54  | 74%   | -0.97 | 74%   | 0.32  | 71%   |
> ||MOD | 0.55 | 0.93 | 60% | 0.96 | *0.92* | *84%* | *0.38* | *78%* | *0.74* | *74%* |
> |                | **Aligner**        | **0.67**  | 1.32  | *72%*   | *0.97*  | 0.63  | 70%   | -1.39 | 12%   | 0.19  | 51%   |
> |                | **Steering**       | 0.63  | 1.17  | 38%   | 0.96  | 0.81  | 66%   | 0.17  | 70%   | 0.72 | 58%   |
> |                | PAD           | *0.65*  | *1.34*  | **74%**   | **0.98**  | **1.03**  | **92%**   | **0.82**  | **86%**   | **1.06**  | **84%**   |
> | HelpSteer dataset | Base       | 0.57  | 0.56  | -     | 0.98  | 0.5   | -     | -0.56 | -     | 0.17  | -     |
> ||Rewarded soups | 0.52 | 0.63 | 46% | 0.90 | 0.66 | 60% | 0.00 | 48% | 0.43 | 51% |
> |                | Preference Prompting | 0.52  | 0.52  | 56%   | 0.84  | *0.96*  | 82%   | -0.49 | *86%*  | 0.33  | *75%*   |
> |                | MetaAligner    | 0.51  | **1.13**  | *58%*   | 0.9   | 0.23  | 82%   | -0.69 | 76%   | 0.22  | 72%   |
> ||MOD | 0.50 | 0.66 | 56% | *0.97* | 0.78 | 70% | *0.24* | 52% | *0.56* | 59% |
> |                | **Aligner**        | *0.59*  | 0.88  | 54%   | **0.99**  | 0.44  | **90%**   | -0.77 | 8%    | 0.18  | 51%   |
> |                | **Steering**       | 0.49  | 0.68  | 40%   | 0.88  | 0.7   | 72%   | 0.18  | 70%   | 0.52  | 61%   |
> |                | PAD           | **0.65**  | *0.89*  | **62%**   | 0.94  | **0.99**  | *86%*   | **1.01**  | **92%**   | **0.96**  | **80%**   |
>
> ---
>
> References
>
> [1] Konen K, Jentzsch S, Diallo D, et al. Style Vectors for Steering Generative Large Language Model[J]. arXiv preprint arXiv:2402.01618, 2024.
>
> [2] Ji J, Chen B, Lou H, et al. Aligner: Achieving efficient alignment through weak-to-strong correction[J]. arXiv preprint arXiv:2402.02416, 2024.
>
> [3] https://github.com/DLR-SC/style-vectors-for-steering-llms
>
> [4] Rewards in-context: Multi-objective alignment of foundation models with dynamic preference adjustment.
>
> [5] https://huggingface.co/aligner/aligner-7b-v1.0

---

> ### Author Response · Authors · 2024-11-19
> **Response to Questions 1-4**
>
> > **Question 1: During the first training phase, is personalized data p included? If so, in what form is it integrated? Could you provide an example?**
>
> Yes, our training data includes personalized preferences. We construct it following the method described in [6] by prepending synthetic user prompts to instructions.
> For example, consider the original instruction x "What is needed for self-sufficient living spaces?", the personalized preference p is the synthetic user prompt "Generate a response that is helpful."
>
> A Prompt Template is provided in Section A.2 (Line 851-858) for better illustration. During both training and testing, x and p serve jointly as inputs to the model, as also depicted in Figure 1 in our manuscript.
>
> ---
>
> > **Question 2: Is there a distinction between the training data used in the two stages?**
>
> The training data remains the same across both stages, but the focus of the training shifts. In the first stage, we aim to optimize the backbone to learn general features. In the second stage, we focus on optimizing the value head to learn different user preferences. For further clarification, we have included an algorithm box in Appendix A.1 (Line 864-890) of the revised manuscript, featuring detailed pseudocode for the practical training process of the personalized reward model.
>
> ---
>
> > **Question 3: Theoretically, PAD appears highly similar to DPO, differing only by the additional hyperparameter β and the personalized parameter Wp. Can the first training stage be understood as DPO, with the second training stage merely training a value head?**
>
> Thank you for your insightful comment. We wish to further clarify the connections and distinctions between our work and DPO.
>
> Broadly speaking, **the primary distinction between PAD and DPO is that DPO optimizes a policy model, whereas PAD trains a personalized reward model.** PAD is a strategy for alignment during the decoding phase and does not require training a policy model. It guides the decoding process to achieve alignment through the optimized personalized reward model.
>
> Specifically, the loss function of our personalized reward model (Eq. 9) bears a formal resemblance to the objective function of DPO. **Inspired by the works on DPO [7,8], we have extended existing research in two significant ways, which constitute the technical novelty of this paper.** First, we formally demonstrate that by rewriting the reward function in the manner of DPO (Eq. 8), we can derive the implicit Q function with the optimized personalized reward model (Eq. 10), making it suitable for guidance during the decoding phase. Secondly, by decoupling personalized preferences from the state in reward modeling, we demonstrate that personalized preferences can be separated from the MDP dynamics (Eq. 4 to Eq. 7), thereby facilitating easier transfer to different user preferences.
>
> In summary, PAD is a decoding-time alignment algorithm that performs decoding under the guidance of a personalized reward model. Within this framework, we incorporate the characteristics of DPO to train the personalized reward model, enabling it to provide token-level guidance. Furthermore, through an innovative personalized reward modeling strategy, we ensure that the personalized rewards are generalizable across different preferences.
>
> We hope this response satisfactorily clarifies the distinctions and connections between PAD and DPO. We appreciate your keen observations and are eager to discuss any further concerns you might have.
>
> ---
>
> > **Question 4: What does π^ref represent in Equation (13)?**
>
> Thank you for pointing out this issue. We recognize that we did not clarify the meaning of $\pi_{ref}$ in this section. In fact, $\pi_{ref}$ here is consistent with earlier mentions (Eq. 3) and represents the reference policy, which is typically used as the initial policy $\pi_{\theta}$ in frameworks such as RLHF or DPO. To be more specific, in this paper, we use Llama-3-8B as the backbone of our personalized reward model. Thus, $\pi_{ref}$ is Llama-3-8B, and $\pi_{\theta}^*$ is the trained personalized reward model.
> Based on your suggestions, we have provided further clarification on this point in the revised manuscript (Line 280-281).
>
> ---
>
> References
>
> [6] Personalized soups: Personalized large language model alignment via post-hoc parameter merging
>
> [7] From r to Q∗: Your Language Model is Secretly a Q-Function
>
> [8] Direct Preference Optimization: Your Language Model is Secretly a Reward Model

---

> > ### Comment · Reviewer_6ie9 · 2024-11-19
> >
> > Thank you for the detailed response. Most of my concerns have been resolved, and I have increased the overall score from 5 to 6.
> >
> > **However, there is still one issue I am struggling to understand.**
> >
> > The output size of the score head in the PAD method matches the vocabulary size of the base model. As far as I know, the vocabulary sizes of different LLMs are not identical. For instance, the vocabulary size of Gemma-2B-it is 256,000, whereas that of Llama3-8B-Instruct is 128,256. How, then, can this method be applied to different base models without retraining?
> >
> > **A truly model-agnostic approach offers broader applicability and greater research value.**

---

> > > ### Author Response · Authors · 2024-11-19
> > > **Reply to the further concern**
> > >
> > > **Thank you for acknowledging our response, and we extend our best wishes to you!**
> > >
> > > ---
> > >
> > > ### Response to the further concern:
> > >
> > > Yes, as you pointed out, in this experiment, we are faced with the challenge of differing vocabularies between the base model (LM) and the reward model (RM). If the vocabularies of the LM and RM are different, we must first decode the text and candidate tokens into natural language using the LM's tokenizer, and then encode using the RM's tokenizer to calculate rewards. The pseudocode is specified as follows:
> > >
> > > *Note: we use "LM" to denote the base model and "RM" to represent the personalized reward model. We use uppercase "S" and "A" for encodings, and lowercase "s" and "a" for text. The terms "encode" and "decode" refer to encoding or decoding process using the LM or RM tokenizer.*
> > >
> > > ```
> > > for t = 0 to T
> > >   S_t = LM.encode(s_t)
> > >   Calculate probability distribution LM(A|S_t) and retain top-K candidates
> > >   S_t' = RM.encode(LM.decode(S_t))
> > >   A' = RM.encode(LM.decode(A))
> > >   Calculate rewards RM(A'|S_t')
> > >   Weighted Score = LM(A|S_t) + β * RM(A'|S_t')
> > >   a_t = LM.decode(argmax_a(Weighted Score))
> > >   s_t+1 = (s_t, a_t)
> > > ```
> > >
> > > Note that when calculating the Weighted Score, the probability and rewards are only for the top-k candidates, not the entire vocabulary. Therefore, issues related to differing vocabulary sizes do not arise. However, the current implementation still has its limitations. Due to differences in vocabulary size, a' may not correspond to the same word as a. In such cases, while the above method ensures that the generated word remains a top-K candidate of the base model, it may lead to inaccurate scoring by the reward model. We believe that this is also why the performance improvements are not as pronounced compared to Llama-3.
> > >
> > > We agree that a truly model-agnostic approach offers broader applicability and greater research value. We acknowledge that, in this paper, we have only empirically demonstrated the efficacy of PAD in a model-agnostic setting, without providing theoretical support for model-agnosticism. We have observed that some recent studies [1,2] are also exploring alignment paradigms that can be broadly applied to various base models. However, they may still encounter similar challenges. We are currently investigating this area and believe that such technology holds significant potential for the future of large language model alignment.
> > >
> > > [1] Weak-to-Strong Search:  Align Large Language Models via  Searching over Small Language Models
> > >
> > > [2] Transferable Post-training via Inverse Value Learning

---

> > > > ### Comment · Reviewer_6ie9 · 2024-11-22
> > > >
> > > > I'm glad that you acknowledged your limitation of considering only the top-k candidates. However, that is somehow a good solution!

---

> > > > > ### Author Response · Authors · 2024-11-22
> > > > > **Thank you very much**
> > > > >
> > > > > **Thank you very much for your kind recognition and the thoughtful insights provided in your review. We deeply appreciate your expert suggestions. We also want to extend our best wishes to you! Thank you once again for your careful consideration and supportive words.**

---

### Official Review · Reviewer_LdS8 · 2024-11-13

**Soundness:** 3
**Presentation:** 3
**Contribution:** 3
**Rating:** 6
**Confidence:** 2

**Summary:**

The paper introduces a novel framework for aligning large language model (LLM) outputs with diverse personalized preferences during the inference phase, eliminating the need for additional training and different models for different preferences. The authors theoretically derive a different formulation of the personalized preference model policy with unique personalized reward modeling strategy, which decouples the text generation process from personalized preferences, allowing for generalizable token-level personalized rewards. This decoupling allows to have a single model trained once and use test-time compute and reward computation at each generation step to achieve the preference stated.

**Strengths:**

1. The paper discusses that PAD requires only a single policy model aligned with general preferences, eliminating additional training. This means that the algorithm operates efficiently without the need for the creation of multiple specialized models. The contrast with previous works in these aspects is detailed very well in Table 1.

2. Great presentation of the theoretical aspects of the algorithm. The experiment section a bit lacking in details though. Yet the analysis of different baselines and different datasets is commendable.

**Weaknesses:**

1. Not much details are shared on how the model can generalize to unseen preferences not seen during the training. The w_p is dependent on p being specified so new preferences would need some retraining.

2. While the experiment section compares the performance of the model against the baselines, the latency and cost is not compared which will have practical implication in real world use.

**Questions:**

see weaknesses

---

> ### Author Response · Authors · 2024-11-19
> **Response to Weaknesses 1-2**
>
> Thank you for your constructive comments. We have carefully considered your suggestions and would like to provide the following clarifications. We have also updated the manuscript, with the modified contents highlighted in orange for your review.
>
> > **Weakness 1: Not much details are shared on how the model can generalize to unseen preferences not seen during the training. The w_p is dependent on p being specified so new preferences would need some retraining.**
>
> Thank you for your insightful comment. "PAD can generalize to unseen preferences" is one of the main contributions of this paper, and we wish to further clarify it.
>
> Firstly, by decoupling personalized preferences from the state in reward modeling, we demonstrate that personalized preferences can be separated from the MDP dynamics (Eq. 4 to Eq. 7), thereby facilitating easier generalization to different user preferences. Then, we provide a theoretical analysis in Section 3.3 on PAD's ability to generalize to unseen preferences. The main intuition is that if PAD has previously aligned to a similar personalized preference, it should generalize well to the new preference.
>
> In addition, we conducted empirical analysis in Section 4.2, Line 416, where we performed experiments on alignment to customized preferences that were not seen during the training phase. The results confirm the superiority of PAD in generalizing to unseen personalized preferences.
>
> In summary, PAD does not require pre-defined personalized preferences during the training phase to generalize to unseen preferences. Should you have any further concerns, please do not hesitate to contact us.
>
> ---
>
> > **Weakness 2: While the experiment section compares the performance of the model against the baselines, the latency and cost is not compared which will have practical implication in real world use.**
>
> Thank you for your insightful comment. Latency and computational overhead are currently focal points in decoding-time alignment research.
>
> Empirically, we have explored the effects of different decoding strategies on latency during the decoding phase in Section 4.3, "Effect of Decoding Strategies" (in the original manuscript).
> The empirical results are reported in Table C3 (Line 1053 in the revised manuscript). The results are also provided in the following table. 'Base' represents conventional greedy decoding of vanilla base models. The time costs for training-based models closely align with those of the 'Base' configuration. 'Greedy' and 'stochastic' represent decoding-time alignment by PAD algorithm. Our results reveal that generation time of PAD increases by 2-3 times compared with conventional greedy decoding. Additionally, we have quantified the memory overhead of PAD. Decoding with PAD (trained from Llama-3-8B) requires an additional 17,095 MB.
>
> **Despite this increase in processing time and memory, there is a notable enhancement in performance across all dimensions (e.g., 26% increase as for helpful and 24% increase as for harmless), indicating a reasonable tradeoff between alignment performance and inference costs.**
> Moreover, PAD's advantage lies in its ability to align LLM outputs with diverse personalized preferences during the inference phase. This approach eliminates the need for the exhaustive process of retraining the RL model and allows for rapid adaptation in response to changes in personalized preferences. In addition, the efficiency gap can be further narrowed by employing a smaller reward model or parallelizing the reward computation across multiple candidates.
>
> In summary, while we acknowledge and discuss the computational overhead introduced by PAD during the inference phase, the reduced training costs and superior performance make this a worthwhile trade-off. Additionally, we suggest that more efficient decoding-time alignment could be achieved using existing strategies.
> **Based on your suggestions, we have added a new paragraph in Section 4.3 (Line 512-519) and Table C3 (Line 1053) that specifically discusses the Computational Cost for PAD.** We appreciate your feedback, which has undoubtedly enhanced both the quality and comprehensiveness of our paper.
>
> **Table. Comparison of the performance and computational  cost of various decoding strategies.** Base: conventional greedy decoding with only the base model. Greedy: select the token with maximum score with PAD. Stochastic: sample a token from the probability distribution of the top-k candidate with PAD.
>
> | Strategy  | Helpful    | Harmless   | Humor     | Diversity | Time (s)  | Memory (MB) |
> |-----------|------------|------------|-----------|-----------|-----------|-------------|
> | Base      | 1.06 | 0.83 | -0.92 | 0.75 | 120 (±1.12)| 16,950      |
> | Greedy    | 1.34 | 1.03 | 0.82  | 0.86 | 256 (±1.94)| 34,045      |
> | Stochastic| 1.33 | 0.93 | 0.92  | 0.87 | 358 (±3.72)| 34,045      |

---

> ### Author Response · Authors · 2024-11-23
>
> Dear Reviewer LdS8,
>
> We hope this message finds you well.
>
> We sincerely apologize for any inconvenience our submission may have caused and greatly appreciate the time and effort you have devoted to reviewing our paper. Your insights and comments have been invaluable to us.
>
> We have carefully considered each of your suggestions and have made comprehensive, point-to-point responses and revisions in our revised manuscript. These adjustments are aimed at thoroughly addressing the concerns you raised, and we hope that our efforts resonate with your expectations for the paper.
>
> At your convenience, we would be grateful if you could review our responses and the corresponding revisions. Should there be any further issues or if additional clarification is needed, please know that we are fully prepared to make the necessary adjustments.
>
> Thank you once again for your dedication and assistance.
>
> Best regards,
>
> The Authors

---

> > ### Comment · Reviewer_LdS8 · 2024-12-01
> > **Response to the rebuttal**
> >
> > I would like to thank the authors for their response. I have adjusted my rating accordingly.
> >
> > It would be great if you can open source the code with the paper.

---

> > > ### Author Response · Authors · 2024-12-01
> > >
> > > Dear Reviewer LdS8,
> > >
> > > Thank you very much for your kind recognition and the thoughtful insights. We deeply appreciate your expert suggestions and are committed to integrating and open-sourcing the code as soon as possible.
> > >
> > > We also want to extend our best wishes to you! Thank you once again for your careful consideration and supportive words. Your feedback is invaluable to us, and we are eager to continue improving our work with your guidance.
> > >
> > > Best regards,
> > >
> > > Authors

---

### Author Response · Authors · 2024-11-22
**Global Response**

Dear reviewers,

**We sincerely appreciate the time and effort you devoted to reviewing our paper. Your detailed feedback has been invaluable, and we have made comprehensive revisions based on your insightful comments.**

We appreciate that the reviewers acknowledge the **advantages** of our work:

1. **Technical Novelty**: Our approach uniquely decouples personalized preferences from the Markov Decision Process and introduces the generation of generalizable token-level personalized rewards (Reviewer NqpU). This innovative concept stands out because it does not require further training of the policy or multiple reward models, which is a common necessity in many existing works (Reviewer NqpU). The paper provides a detailed contrast with previous work, highlighting the novel aspects of our methodology (Reviewer LdS8).

2. **Empirical effectiveness**: The combination of theoretical innovation and empirical evidence significantly reinforces the effectiveness of our method. The paper conducts extensive ablation studies to explore PAD’s robustness and its capability to generalize across different conditions, underscoring the practical utility of our approach (Reviewer 6ie9).

3. **Clear presentation**: The paper is well-received for its excellent presentation of the theoretical aspects of the algorithm (Reviewer LdS8). It is well-written, organized, and easy to follow, effectively building up the concept of PAD (Reviewer NqpU). The methods section, in particular, is noted for its clarity and formalization, making complex ideas accessible and understandable (Reviewer s7Gt).

On the other hand, we have diligently addressed all the **concerns** by conducting extensive experiments and providing detailed clarifications. Additional experiments and clarifications have been incorporated into the revised manuscript. We have made substantial updates to address the concerns:

1. **Enhanced Experimental Results:**

 - **Computational Cost Analysis** (Reviewers LdS8 and NqpU): Added a new paragraph in Section 4.3 specifically discussing the computational cost associated with PAD.
-  **Expanded Baseline Comparative Analysis** (Reviewer 6ie9): Included two additional baselines in the revised manuscript's Table 2 to enhance comparative analysis.
-  **Pareto Front Analysis** (Reviewer NqpU): Introduced a new Section C.1 discussing the Pareto front analysis on the trade-offs between different preferences for PAD.
- **Expanded Base Model Comparison** (Reviewer NqpU): Added comparisons of PAD performance across various base models to demonstrate its applicability and effectiveness.
- **Robustness Analysis** (Reviewer NqpU): Conducted new experiments on the robustness of PAD to variations in preference expressions, underscoring its stability.
-  **Case Studies** (Reviewer s7Gt): Included case studies illustrating how our method can guide the generation of responses in various directions.

2. **Enriched Implementation Details:**
- **Detailed implementation Details** (Reviewers 6ie9 and NqpU): Added two algorithm boxes that provide detailed pseudocode for the practical implementations of PAD, enhancing the manuscript's clarity.
- **Extended Experimental Explanations** (Reviewer 6ie9): Provided additional explanations of the motivation and experimental setup in Section 4.4.
- **Personalized Prompt Template Disclosure** (Reviewers 6ie9 and s7Gt): Provided the prompt template to elucidate how the prompts for personalized preferences are constructed.

**We hope that these revisions adequately address your concerns. Since the discussion phase is concluding, we would be grateful if you would kindly let us know of any other concerns and if we could further assist in clarifying any other issues.**

---

### Meta-Review · Area_Chair_ksBw · 2024-12-12

**Metareview:**

This paper proposes Personalized Alignment at Decoding-time (PAD) which aims to to align LLM outputs with personalized preferences during inference. The reviewers generally found the proposed personalized reward modeling strategy to be novel and this decoding stage alignment problem to be interesting and important. The authors have well addressed the reviewers' concerns in the rebuttal.

A minor note: The authors should update their references with the conference proceeding version (if accepted) instead of the arXiv version.
For example:

@inproceedings{mudgalcontrolled,
  title={Controlled Decoding from Language Models},
  author={Mudgal, Sidharth and Lee, Jong and Ganapathy, Harish and Li, YaGuang and Wang, Tao and Huang, Yanping and Chen, Zhifeng and Cheng, Heng-Tze and Collins, Michael and Strohman, Trevor and others},
  booktitle={Forty-first International Conference on Machine Learning},
year={2024},
}

@inproceedings{meng2024simpo,
title={Sim{PO}: Simple Preference Optimization with a Reference-Free Reward},
author={Yu Meng and Mengzhou Xia and Danqi Chen},
booktitle={The Thirty-eighth Annual Conference on Neural Information Processing Systems},
year={2024},
}

**Additional Comments On Reviewer Discussion:**

Reviewers reached a consensus on acceptance.

---

### Decision · Program_Chairs · 2025-01-22

Accept (Poster)